# Whole-genome sequencing reveals genomic signatures associated with the inflammatory microenvironments in Chinese NSCLC patients

Cheng Wang[1,2,3], Rong Yin[4], Juncheng Dai[1,2], Yayun Gu[1,2], Shaohua Cui[5], Hongxia Ma[1,2], Zhihong Zhang[6], Jiaqi Huang[7], Na Qin[1,2], Tao Jiang[1,2], Liguo Geng[1,2], Meng Zhu[1,2], Zhening Pu[1,2], Fangzhi Du[1,2], Yuzhuo Wang[1,2], Jianshui Yang[1,2], Liang Chen[8], Qianghu Wang[3], Yue Jiang[1,2], Lili Dong[5], Yihong Yao[7], Guangfu Jin [1,2], Zhibin Hu[1,2], Liyan Jiang[5], Lin Xu[4] & Hongbing Shen[1,2]

Chinese lung cancer patients have distinct epidemiologic and genomic features, highlighting the presence of specific etiologic mechanisms other than smoking. Here, we present a comprehensive genomic landscape of 149 non-small cell lung cancer (NSCLC) cases and identify 15 potential driver genes. We reveal that Chinese patients are specially characterized by not only highly clustered *EGFR* mutations but a mutational signature (MS3, 33.7%), that is associated with inflammatory tumor-infiltrating B lymphocytes ($P = 0.001$). The *EGFR* mutation rate is significantly increased with the proportion of the MS3 signature ($P = 9.37 \times 10^{-5}$). TCGA data confirm that the infiltrating B lymphocyte abundance is significantly higher in the *EGFR*-mutated patients ($P = 0.007$). Additionally, MS3-high patients carry a higher contribution of distant chromosomal rearrangements >1 Mb ($P = 1.35 \times 10^{-7}$), some of which result in fusions involving genes with important functions (i.e., *ALK* and *RET*). Thus, inflammatory infiltration may contribute to the accumulation of *EGFR* mutations, especially in never-smokers.

[1] Department of Epidemiology and Biostatistics, School of Public Health, Nanjing Medical University, 211116 Nanjing, China. [2] Jiangsu Key Lab of Cancer Biomarkers, Prevention and Treatment, Jiangsu Collaborative Innovation Center for Cancer Personalized Medicine, Nanjing Medical University, 211116 Nanjing, China. [3] Department of Bioinformatics, School of Biomedical Engineering and Informatics, Nanjing Medical University, 211116 Nanjing, China. [4] Department of Thoracic Surgery, Jiangsu Cancer Hospital, Jiangsu Institute of Cancer Research, Nanjing Medical University Affiliated Cancer Hospital, Jiangsu Key Laboratory of Molecular and Translational Cancer Research, 210029 Nanjing, China. [5] Department of Respiratory Medicine, Shanghai Chest Hospital, Shanghai Jiao Tong University, 200030 Shanghai, China. [6] Department of Pathology, First Affiliated Hospital, Nanjing Medical University, 210029 Nanjing, China. [7] Cellular Biomedicine Group, Inc., 200233 Shanghai, China. [8] Department of Thoracic Surgery, First Affiliated Hospital, Nanjing Medical University, 210029 Nanjing, China. These authors contributed equally: Cheng Wang, Rong Yin, Juncheng Dai, Yayun Gu, Shaohua Cui, Hongxia Ma. These authors jointly supervised this work: Hongbing Shen, Lin Xu, Liyan Jiang, Zhibin Hu. Correspondence and requests for materials should be addressed to L.J. (email: Jiang_liyan2000@126.com) or to L.X. (email: xulin83@vip.sina.com) or to H.S. (email: hbshen@njmu.edu.cn)

Lung cancer is the most commonly diagnosed cancer and the leading cause of cancer deaths both globally and in China[1]. Non-small cell lung cancer (NSCLC) is the major histological type of lung cancer and mainly includes adenocarcinoma (AD) and squamous cell carcinoma (SCC). Over the past decades, next-generation sequencing has accelerated the systematic characterization of genomic events, including single base substitutions and small insertions/deletions (indels), and has yielded substantial insights into the unique and shared genomic features of the two NSCLC subtypes[2–4].

A series of candidate "driver" genes involved in the receptor tyrosine kinase (RTK) pathway were identified in recent studies. Alterations in some genes, including *EGFR*, *KRAS*, and *PIK3CA*, have been shown to trigger carcinogenesis of lung cancer in the mouse models[5].

Genomic studies of lung cancer have mainly been conducted in patients from Western countries[2–4]; however, Chinese patients have unique epidemiological features, especially women. For example, the lung cancer rates are higher in Chinese women than among women in some European countries despite an extremely low prevalence of smoking[1], indicating the presence of other carcinogens and carcinogenic mechanisms. The inverse mutation rates of *EGFR* and *KRAS* in Chinese NSCLC patients[6] have provided some clues for the etiologic mechanisms, but the full picture remains unclear.

Carcinogens commonly leave imprints in the DNA. Thus, signatures of genomic events such as mutational signatures (MSs) are widely used in cancer etiological studies[7–9]. Whole-genome sequencing (WGS) is a powerful technology for investigations of alterations in intronic and intergenic regions as well as in the exome, and provides an opportunity to delineate the complete MSs imprinted on the genome during the mutagenic process[7–11]. Moreover, WGS enables the detection of genomic rearrangements that are major but less-studied components of structural alterations in cancer genomics[10,11].

In this study, we analyze genomic events in 149 NSCLC cases (92 WGS and 57 whole-exome sequencing (WES)) coupled with The Cancer Genome Atlas (TCGA) WGS data from 100 NSCLC patients and comprehensively identified the distinct genomic features of Chinese NSCLC patients. We show that the differences (i.e., the diverse mutation rates of *EGFR*) are mainly attributed to a subtype of patients with enriched inflammatory tumor-infiltrating lymphocytes (TILs). These patients are characterized by a distinctive MS and large-scale rearrangements.

## Results

**Burden of genomic events in Chinese NSCLC patients**. To identify somatic alterations in Chinese NSCLC patients, we performed WGS on tumor-blood pairs from 92 cases (57 lung AD and 35 lung SCC) and WES on additional 57 pairs (27 lung AD and 30 lung SCC) from Nanjing Lung Cancer Cohort (NJLCC, Supplementary Fig. 1). We observed a lower smoking rate in the Chinese NSCLC patients (53.7%) than in the TCGA patients (85.5%) (Supplementary Table 1), which was most obvious in lung AD patients (NJLCC: 34.5% vs. TCGA: 82.4%). Almost all of the female patients enrolled in this NJLCC cohort were lifelong non-smokers, with typical demographic and clinical characteristics for the disease (Supplementary Table 1).

Unlike comparable mutation rates between lung AD and SCC in previous studies of other populations ($\text{Median}_{\text{AD}} = 8.7$ mutations/Mb and $\text{Median}_{\text{SCC}} = 9.7$ mutations/Mb)[2], we observed polarized mutation rates in the Chinese patients ($\text{Median}_{\text{AD}} = 2.17$ mutations/Mb, and $\text{Median}_{\text{SCC}} = 13.64$ mutations/Mb, Wilcoxon's rank sum test $P = 3.45 \times 10^{-8}$, Fig. 1a). Similar results were observed in the patients following WES

(Supplementary Fig. 2a). Additionally, we detected abundant genomic structural rearrangements in these patients, with a median of 126 rearrangements (range 16–405, Fig. 1a). However, we observed no significant differences between patients with varied histologic or smoking statuses.

**Diverse mutation patterns of frequently altered genes**. To determine the most common cancer genes involved in Chinese NSCLC patients, we combined somatic substitutions and indels in protein-coding exons with data obtained from both WGS and WES. Overall, we identified 27 significantly mutated genes (SMGs) in these NSCLC patients using the IntOGen and Mut-SigCV frameworks (Supplementary Table 2), of which 15 genes were potential driver genes in previous studies[2,12,13] (Fig. 1b). Three additional driver candidates were identified in a histological subtype analysis (the chromatin remodeling gene *SMARCA4* in AD and *FAT1* and *SVEP1* in SCC, Supplementary Tables 3 and 4). Furthermore, we also identified 30 frequently altered regions in our patients (Supplementary Fig. 2b and Supplementary Table 5). All of these regions overlapped with previously identified regions[2] but deletions at 6p21.32 and 9p13.1. The peak region of deletions at 6p21.32 included major histocompatibility complex (MHC) class II molecules (Supplementary Fig. 2b).

Seven genes were mutated with different rates between the AD and SCC patients. Two oncogenes (*EGFR* and *KRAS*) in the RTK pathways were mutated more frequently in AD. Notably, we observed highly accumulated *EGFR* mutations in our patients (31.5%), and the *EGFR* mutation rate was significantly higher in our AD patients (52.0%) than in the patients from TCGA project (13.6%) (Fisher's exact test $P = 6.69 \times 10^{-14}$, Fig. 1c). *EGFR* amplifications also occurred frequently in our patients (18.8%) and most of which (10.7%) co-occurred with the *EGFR* mutations. In contrast to *KRAS*, *EGFR* was also mutated in the SCC patients (4.6%) and the frequency was higher than the frequency in the SCC patients from TCGA (3.4%), while the smoking rate was relatively lower in our SCC patients (82.9%) than in TCGA SCC patients (93.8%). Smoking-related *KRAS* mutations were less common in our patients (8.0%) than in TCGA patients (31.3%) and occurred exclusively with *EGFR* mutations. *STK11*, which is another gene exclusively mutated with *EGFR*[14], also had a lower frequency in our AD patients (Fig. 1c). Moreover, we observed a higher rate of *RB1* mutations in AD than in SCC, though the difference was not significant, and almost all of the *RB1* mutations co-occurred with *TP53* mutations (Fig. 1b). In addition, *RB1* mutations occurred more frequently in our AD patients (15.5%) than in TCGA patients (5.4%).

Most of the SCC-specific driver genes were classic tumor suppressors, including *TP53* (75%), *KMT2D* (28%), *CDKN2A* (22%), and *FBXW7* (9%), indicating the highly disorganized status of SCC cells. Compared with TCGA SCC patients, our SCC patients had more frameshift indels in *CDKN2A*, but there was no difference between the mutation rates (Fig. 1c); mutations in oncogenic *NFE2L2* occurred only in the SCC patients and exhibited a significantly higher frequency in our SCC patients, whereas mutations in *KEAP1* and *CUL3* (*KEAP1-NRF2* pathway) were so rare that they could not be identified as SMGs.

**Three MSs in Chinese patients**. To illuminate the etiologic mechanisms of the diverse mutation rates in Chinese patients and decipher the mechanisms underlying the mutagenic and tumorigenic processes of lung cancer, we adopted a non-negative matrix factorization (NMF) to extract MSs from 96 subtypes of three-base context of mutations. Three prominent signatures were detected (Supplementary Fig. 3a): MS1, which is characterized by C > T mutations at the TpC dinucleotide and an

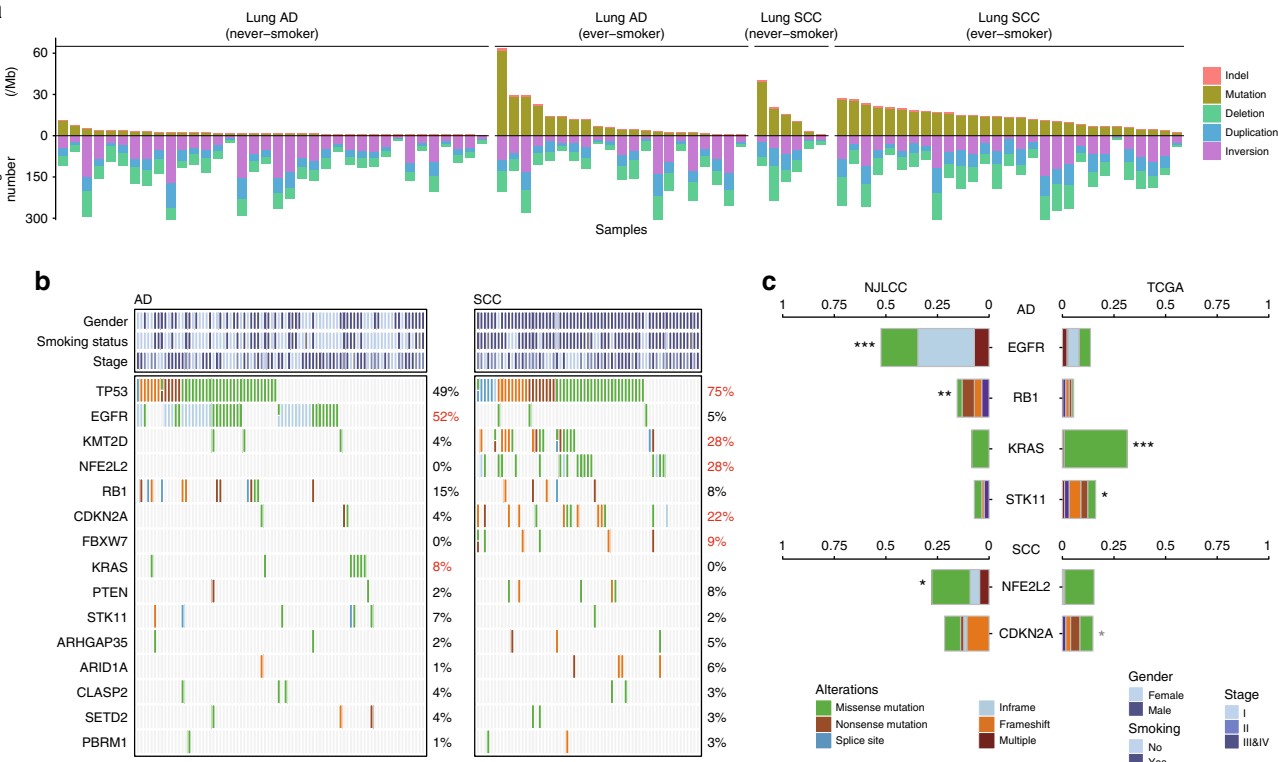

**Fig. 1** Distinct mutation patterns in Chinese NSCLC patients. **a** Loads of somatic mutations (substitutions and indels) or intra-chromosomal rearrangements (deletions, duplications, and inversions) identified in whole-genome sequencing data from Chinese NSCLC patients; **b** Mutation landscapes of significantly mutated genes (SMGs) in Chinese NSCLC patients. The frequencies shown in red indicate genes with significant differences between AD and SCC (FDR $q < 0.05$); **c** Significantly mutated genes (SMGs) with significantly different alteration frequencies in patients from the NJLCC and TCGA. Fisher's exact test, ***$P < 0.001$; **$P < 0.01$; *$P < 0.05$; the gray star indicates a significantly different constitution of mutation types

APOBEC-driven hyper-mutated phenotype (Supplementary Fig. 3b); MS2, which is characterized mainly by C > A mutations with smaller contributions from other base substitution classes and a complex pattern caused by exposure to tobacco carcinogens (Supplementary Fig. 3b); and MS3, which is characterized by mutations distributed across all 96 subtypes of base substitutions with a predominance of C > T and T > C mutations, although the underlying mechanisms of MS3 are not well understood[7] (Supplementary Fig. 3b). Similar signatures were extracted from 100 WGS data sets from TCGA NSCLC patients (Supplementary Fig. 4a).

In our patients, APOBEC-related MS1-dominant patients were only observed in a small number of AD patients (10.9%), although the MS1 proportion showed no significant difference between AD and SCC (Supplementary Fig. 3c). The smoking-related MS2 was overwhelmingly seen in SCC, and every patient carried MS2 mutations (Supplementary Fig. 3c). Similar MS2 proportion was observed in the SCC patients from TCGA (Supplementary Fig. 3c & 4b).

In contrast to MS2, MS3 was a predominant signature in AD (Wilcoxon's rank sum test $P = 5.77 \times 10^{-5}$), never-smokers (Wilcoxon's rank sum test $P = 3.76 \times 10^{-6}$), and female patients (Wilcoxon's rank sum test $P = 4.09 \times 10^{-3}$) (Supplementary Fig. 3c). Moreover, 33.7% patients (31/92) were defined as MS3 dominant in our patients. The proportion was more than twice as much as in TCGA patients (Fisher's exact test OR = 2.30, $P = 0.01$), suggesting that MS3 was mainly present in the Chinese patients.

**Characterized genomic alterations in MS3-dominant patients.** Next, we investigated whether MS3 signature was associated with Chinese-specific *EGFR* mutations. Distinct from the dispersed

mutation pattern in TCGA patients, the *EGFR* mutations in our patients were not only more frequent but were also highly recurrent (Fig. 2a). A total of 39 patients carried mutations sensitive to *EGFR* tyrosine kinase inhibitors (TKI) (19th exon deletion: 20 and L858R: 19), and five patients carried TKI-resistant mutations (insertions on the 20th exon and the T790M mutation). One patient carried both TKI-sensitive and TKI-resistant mutations (Fig. 2a). As expected, the *EGFR* mutations mainly occurred in the MS3-dominant NSCLC patients (Cochran–Armitage trend test $P = 9.37 \times 10^{-5}$, Fig. 2b). The mutation rate increased with the increasing MS3 proportion and was maximized in the 7th and 8th MS3 groups in our patients (Fig. 2c). The *EGFR* mutation rate in the patients in these groups reached 80% (Fig. 2c). Using the same MS3 cutoffs, we found that the two groups had the fewest patients in the TCGA data, which explained the differences in the *EGFR* mutation rates between our patients and TCGA patients (Fig. 2c).

We also noted that the patients in the highest MS3 quantile carried much fewer *EGFR* mutations (2/10) but carried a specific copy number profile (GISTIC cluster 1, Fig. 2b) clustered from the copy number status of 30 highly recurrent amplified/deleted regions, suggesting that structural alterations might also participate in the carcinogenic process in the patients with the highest MS3 signal. Thus, we systematically investigated the fusion genes in our patients. We observed six known fusions in our patients involving *ALK*, *ROS1*, and *RET*. Half of these fusions occurred in the patients in the highest MS3 quantile (Fisher's exact test OR = 10.43, $P = 0.02$, Fig. 2c). Similar results were observed in TCGA data (Fisher's exact test OR = 30.41, $P = 0.004$, Fig. 2c). Moreover, we identified a new fusion transcript involving the ERBB family-related lncRNA *BCAR4*[15] (*CD63-BCAR4*, Supplementary

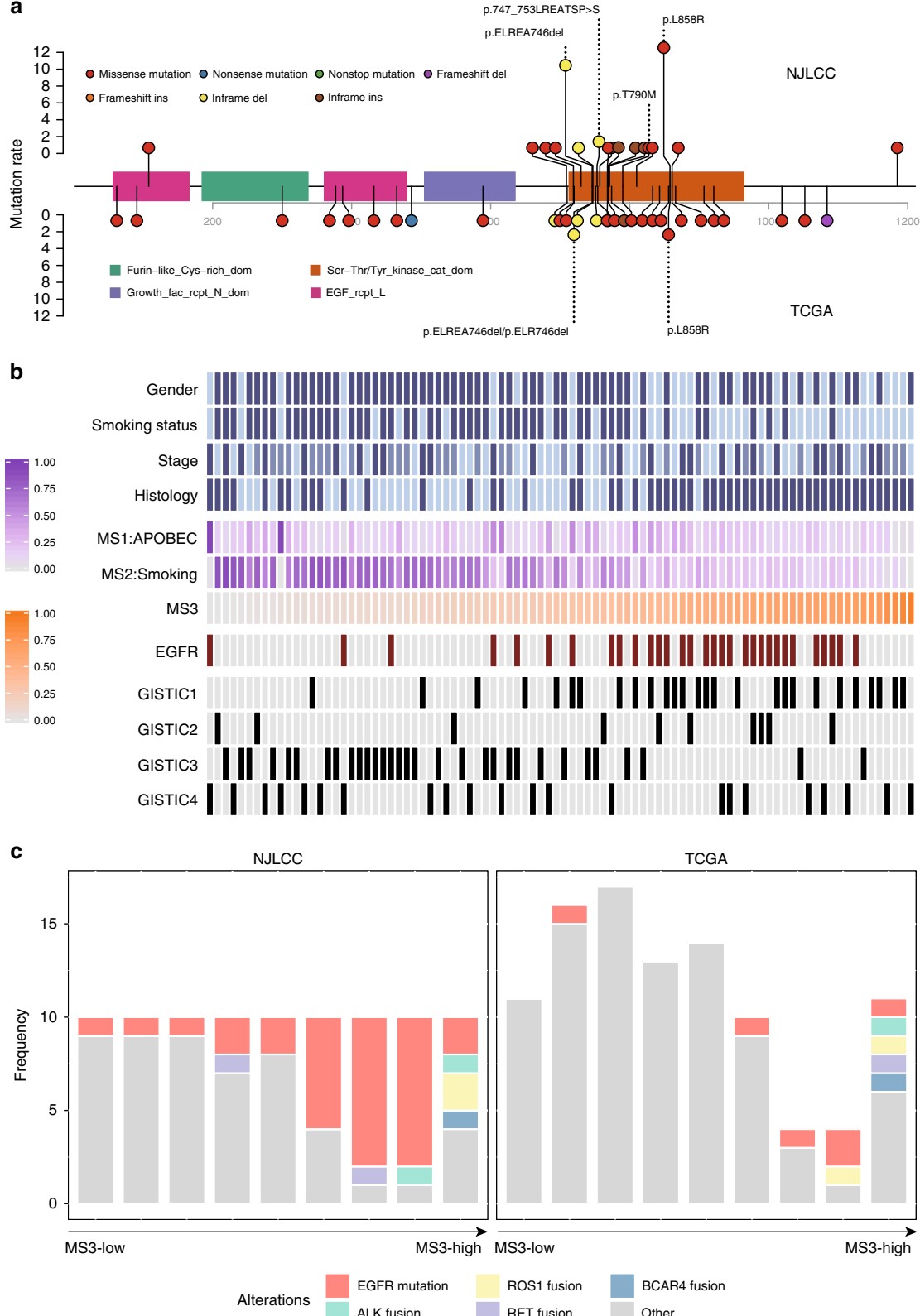

**Fig. 2** Mutation pattern of *EGFR* in NSCLC patients. **a** Lolliplot of *EGFR* mutation rate in the patients from the NJLCC and TCGA. **b** *EGFR* mutations and GISTIC cluster 1 copy number alterations occurred more frequently in the MS3-high patients. **c** The 7th and 8th groups of our NSCLC patients carried the highest *EGFR* mutation rate, whereas much fewer TCGA patients were on the same scale. The genes in RTK pathway were commonly fused in patients with the highest level of the MS3 signal

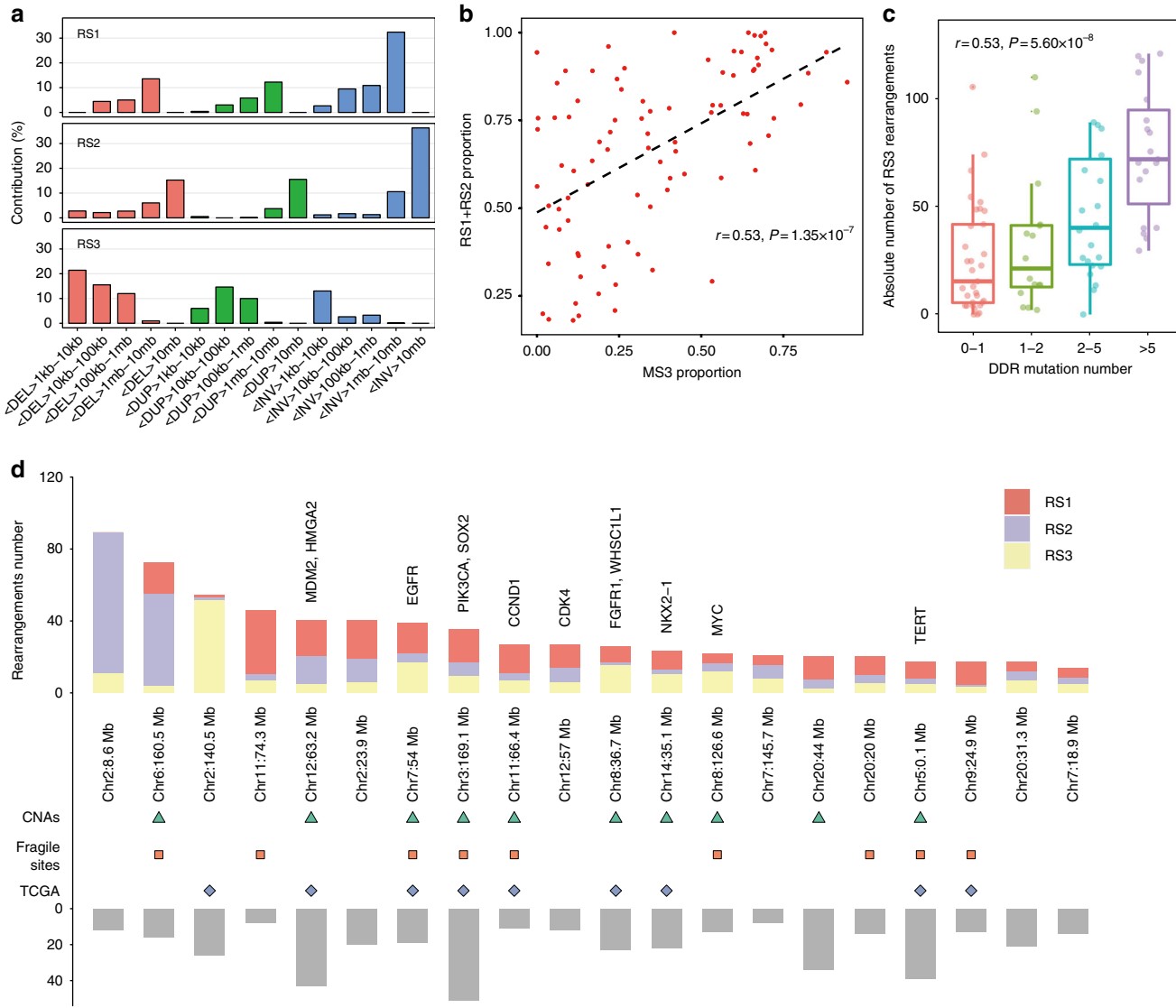

**Fig. 3** Structural rearrangement signatures identified in our patients. **a** Structural rearrangement signatures identified in Chinese NSCLC patients. **b** The proportion of RS1/RS2 was significantly associated with the proportion of MS3. **c** Absolute number of RS3 was associated with functional mutations in DDR pathways. The box plot displays the first and third quartiles (top and bottom of the boxes), the median (band inside the boxes), and the lowest and highest point within 1.5 times the interquartile range of the lower and higher quartile (whiskers). **d** Rearrangement hotspots identified in NSCLC patients. Regions overlapped with copy number altered regions identified in this study were marked as green triangle. Regions overlapped with known fragile sites were marked as red square. Regions overlapped with copy number altered regions identified in the TCGA NSCLC data were marked as blue diamond

Fig. 5a). The expression levels of both this fused gene and *ERBB3* were highly activated in the patient (Supplementary Fig. 5b & c). A *BCAR4* fusion was also observed in a TCGA patient in whom *BCAR4* was directly fused with *ERBB3* to activate the expression of both genes[16] (Supplementary Fig. 5a). Interestingly, the new fusions were only observed in the patients with the highest MS3 signal (Fig. 2c). For patients with enough adjacent normal tissues, we also examined the rearrangements in the adjacent normal tissues, but none of the rearrangements was observed (Supplementary Fig. 6), suggesting that the rearrangements only occurred in tumor tissues.

**MS3 co-occurred with distant rearrangements (>1 Mb).** We also investigated genomic rearrangements in the NSCLC patients. To conduct this analysis, we adopted a rearrangement classification that incorporated 15 subclasses according to a previous study[11]. Application of the NMF algorithm revealed three rearrangement signatures (RSs).

RS1 and RS2 were mainly characterized by rearrangements >1 Mb (Fig. 3a). More than 60% of the intra-chromosomal rearrangements in our patients were classified as RS1/RS2 dominant. We observed a significant correlation between RS1/RS2 and the MS3 proportion (Spearman Correlation $r = 0.53$, $P = 1.35 \times 10^{-7}$, Fig. 3b, Supplementary Fig. 7a). RS1 was incomplete in TCGA NSCLC patients, possibly due to the lower number of MS3-related patients and more smokers (Supplementary Fig. 8).

RS3 was characterized by rearrangements <1 Mb (Fig. 3a). This signature was significantly associated with the smoking-related MS2 (Spearman Correlation $r = 0.52$, $P = 2.22 \times 10^{-7}$) and was also extracted from TCGA NSCLC data (Supplementary Fig. 8). RS3 was once reported to be associated with alterations in the DNA damage response (DDR) gene *BRCA2* in breast cancer patients[11], which suggested that smoking might lead to *BRCA2*-like rearrangements. We observed only four individuals with functional mutations in *BRCA1/BRCA2*, but the RS3

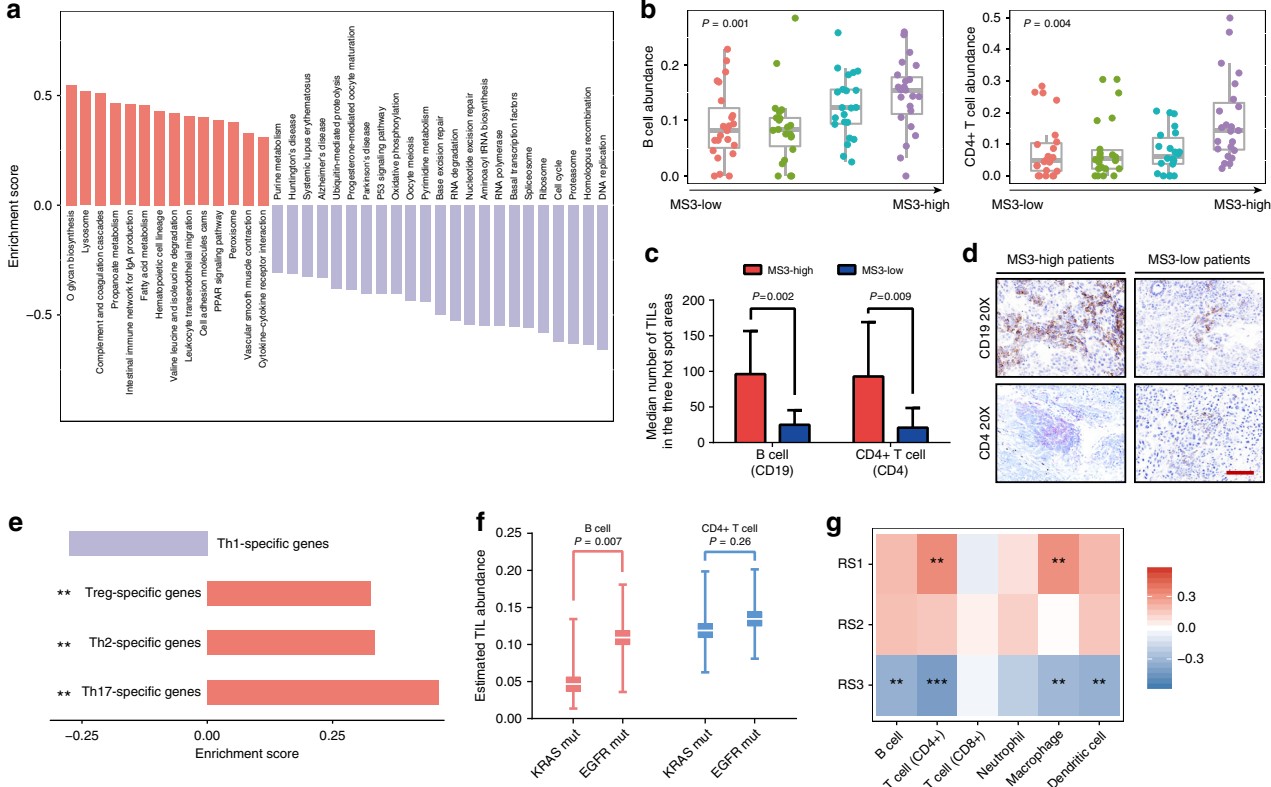

**Fig. 4** MS3 dominance was characterized by enriched tumor-infiltrating lymphocytes (TILs). **a** The GSEA analysis revealed that the genes positively correlated with MS3 were enriched in immunological pathways. **b** B cells and T cells were significantly associated with the MS3 proportion. The box plot displays the first and third quartiles (top and bottom of the boxes), the median (band inside the boxes), and the lowest and highest point within 1.5 times the interquartile range of the lower and higher quartile (whiskers). **c** Results of the immunohistochemistry (IHC) validation of the association (CD19 represents B cells and CD4 represents CD4+ T cells). **d** Representative images of IHC in MS3-high patients and MS3-low patients, scale bar, 100 μm. **e** The GSEA analysis revealed that genes with expression levels correlated with MS3 were significantly enriched in regulatory T cell (Treg)-, T helper cell (Th) 2- and Th17-specific gene sets. **f** B cells were present in significantly higher levels in TCGA patients with *EGFR* mutations than with *KRAS* mutations. The median, first and third quartiles are presented. **g** RS1 contribution was positively correlated with abundance of infiltrating CD4 T cells and macrophages. RS3 contribution was negatively correlated with abundance of multiple TILs. Spearman Correlation ***$P < 0.001$; **$P < 0.01$; *$P < 0.05$

rearrangement number increased with the number of functional mutations in DDR-related genes[17] (Fig. 3c), indicating that other DDR mechanisms were largely responsible for this signature in lung cancer. Additional analysis revealed that the RS3 proportion was significantly associated with mutations in DDR-related genes (Wilcoxon's rank sum test $P = 0.02$) and involved multiple DDR-related pathways (Supplementary Fig. 7b).

Next, we applied a piecewise constant fitting (PCF) approach to find breakpoint hotspots with clustered rearrangements by adjusting the background features[18]. In total, we identified 22 hotspots. Two hotspots at 2p22.3 (*LINC00486*) and 2p25.3 (*TPO-PXDN*) showed an extremely high rearrangement rate (>100/Mb), which might include new fragile sites in NSCLC cells (Supplementary Fig. 7c). Among the remaining 20 hotspots, most of which (17/20) were dominated by RS1/RS2 (Fig. 3d) and 10 hotspots overlapped with the regions with highly altered copy number detected in this study, including 5p15.33 (*TERT*), 3q26.32 (*SOX2* and *PIK3CA*), and 7p11.2 (*EGFR*) (Fig. 3d). Chromosome 2 seems to be vulnerable in lung cancer, since five hotspots were identified in our patients, including highly rearranged hotspots at 2p22.3 and 2p25.3.

**Enriched infiltrating lymphocytes in MS3-dominant patients.** To investigate the potential etiologic mechanism of MS3, we conducted a gene set enrichment analysis (GSEA) to compare the genes expressed according to the proportion of the MS3 signal to

the gene list from 165 KEGG pathways. A total of 14 pathways were significantly enriched with genes positively associated with MS3 (GSEA FDR $q < 0.05$, Fig. 4a), most of which were related to the immune response (i.e., glycan biosynthesis[19], lysosome[20], and intestinal immune network for IgA production).

Tumor-prompting inflammation is one of two major enabling hallmarks of cancer[21] and is widely linked to genomic instability[22], but no inflammatory signal in genomics has ever been described. Thus, we evaluated the association between tumor-prompting inflammation environment and the MS3 signal using in silico TIL estimates from the RNA-seq data. We found a significant positive correlation between the MS3 proportion and multiple TILs (B cells and CD4+ T cells, FDR $q < 0.05$, Fig. 4b and Supplementary Fig. 9a). The immunohistochemistry analysis also confirmed that B cells (CD19) and CD4+ T cells (CD4) were present in significantly higher levels in the MS3-dominant patients compared to the other patients (Fig. 4c, d). Infiltrating B cells are known for their tumor-inducing roles[23], but CD4+ T cells consist of different subpopulations. An additional GSEA analysis also revealed that genes with expression levels correlated with MS3 were significantly enriched in tumor-stimulating regulatory T cell (Treg)-, T helper cell (Th) 2-, and Th17-specific genes (GSEA $P_{Treg} = 0.004$, $P_{Th2} = 0.004$, and $P_{Th17} = 0.008$, Fig. 4e, Supplementary Fig. 9b) but not in Th1-specific genes ($P_{Th1} = 0.43$, Fig. 4e, Supplementary Fig. 9b)[22,23]. T cell subpopulation-specific gene sets were determined by RNA-seq data obtained from purified T lymphocyte cells[24]. These tumor-

prompting T cells commonly exist in a chronic inflammation environment and can lead to the inhibition of specific immune responses[23]. Because the abundance of B cells and CD4+ T cells also varied across NSCLC patients with different histological subtypes (Supplementary Fig. 9c), we used a partial correlation to exclude the immunological differences caused by histological heterogeneity. In our data, the correlation between B cells and MS3 remained significant after adjusting for the histological types (Spearman rank partial correlation $r = 0.21$, $P = 0.047$, Supplementary Table 6). Because the *EGFR* mutations primarily occurred in the MS3-dominant patients and the *KRAS* mutations occurred in the MS2-dominant patients, we compared the TIL levels between patients with the *EGFR* and *KRAS* mutations. We observed a significantly higher B cell abundance in the *EGFR*-mutated NSCLC patients from TCGA (Wilcoxon's rank sum test $P = 0.007$, Fig. 4f) further connecting an inflammatory microenvironments with accumulated *EGFR* mutations.

Similarly, the RS1 proportion was associated with multiple TILs (Fig. 4g). In contrast to the connection between MS3 and B cells, the RS1 signal was associated with another antigen-presenting cell type (macrophage; Spearman Correlation $r = 0.32$, $P = 0.002$) instead of B cells (Spearman Correlation $r = 0.19$, $P = 0.08$). This finding suggested that a different immunologic mechanism was involved in the generation of the long structural alterations.

Notably, most of the depleted pathways in MS3-GSEA analysis served as enriched pathways in smoking-related MS2-GSEA analysis and involved the DNA replication and damage response processes (Fig. 4a and Supplementary Fig. 9d). This finding further supported that smoking-like genomic events were mainly attributed to the DDR process. Thus, tobacco smoking commonly results in thousands of mutations and short genomic rearrangements and results in a heavy burden of genomic alterations, some of which can act as "neo-antigens" and trigger a tumor-eliminating immunological process to inhibit tumor cells[25]. Here, we observed that *CD274* (PD-L1) expression also increased concomitant with the MS2 mutations (Spearman Correlation $r = 0.16$, $P = 0.04$) and the total mutation burden (Spearman Correlation $r = 0.16$, $P = 0.03$).

## Discussion

Tumor-prompting inflammation is defined as a tumor-enabling hallmark of cancer, that is also reported to cause genomic instability and induce genomic alterations, including the MS3-like C > T and T > C mutations[26] and oxidative stress that generates genomic rearrangements[27]. According to a specific MS, our results revealed that Chinese NSCLC patients were enriched with inflammatory TILs (B and CD4+ T lymphocytes)[28]. A consistent immunologic signature was also identified in early-stage lung AD by a very recent study[29], which integrated mass cytometry by time-of-flight (CyTOF), single-cell transcriptomics and multiplex tissue imaging of the lung tumor. The results indicated that the immunologic microenvironments occurred in the initial stage of lung cancer and therefore emphasized their essential role in the development of lung cancer. Recent studies on Chinese lung cancer patients also provided evidence for the association between chronic inflammation and non-smoking lung cancer: Shiels et al. found that circulating inflammation markers were elevated in the lung cancer patients several years before the diagnosis[30]; our recent work also suggested that circulating polyunsaturated fatty acids, which is important participants in the stimulation of chronic inflammation, were causal risk factors of lung cancer, especially for female never-smokers[31]. Our results further identified the association between inflammatory TILs and well-known *EGFR* mutations in Chinese NSCLC patients. In

addition, we reanalyzed sequencing data from a recent study on lung AD and precancerous tissues (atypical adenomatous hyperplasia, AAH)[32]. We found that inflammatory micro-environments formed earlier than the occurrence of *EGFR* mutations (Supplementary Fig. 10). Because inflammatory microenvironments release signaling molecules such as the tumor growth factor EGF[21], which can serve as a ligand for the EGFR protein, we proposed that the microenvironments may select tumor cells with an EGFR protein that is highly activated by functional mutations in *EGFR*, which provides a new potential explanation for the frequent and recurrent *EGFR* mutations in never-smokers. Thus, patients with *EGFR* mutations may benefit from a combination of anti-tumor immunologic therapy and an EGFR inhibitor, especially TKI-resistant individuals, although the causal relationship between TILs and *EGFR* mutations needs further evaluation. Furthermore, we found that such alterations of immunologic microenvironments can be discriminated by intra-rearrangements longer than 1 Mb. The high accumulation of these rearrangements can generate important fused genes involving RTK pathway (i.e., *ALK* and *RET*), that can directly drive the carcinogenic process. These findings illuminated the connection between specific genomic features and tumor-prompting inflammation in NSCLC and emphasized the importance of immunologic alterations in Chinese NSCLC patients.

In contrast to the inflammatory infiltrating NSCLC, smoking-related NSCLC was characterized by driver mutations in tumor suppressors and high level of mutations and short genomic rearrangements, possibly due to the loss of the DDR. Because high mutation load has been linked with cytolytic activity[33,34] and elevated expression of *CD274* (PD-L1) is observed in our smoking-related NSCLC patients, the application of immune checkpoint blockade therapies may be expected in the future. In addition, further studies are warranted to evaluate the response of PARP inhibitor in these patients because of the high proportion BRCA2-like genomic rearrangements, although *BRCA1/2* mutations are seldom observed in NSCLC patients[35].

## Methods

**Sample collection and DNA/RNA extraction**. The study and its design were approved by local ethics committee (Nanjing Medical University and Shanghai Chest Hospital) and all participants provided written informed consent for the research. Patients who received any treatment or neoadjuvant therapy before surgery/biopsy were excluded. Samples (tumor specimens, adjacent normal tissues, and peripheral blood) were obtained during surgical resection. All tissue samples were snap-frozen. HE-stained sections from each sample were subjected to an independent pathology review to confirm that the tumor specimen was histologically consistent with NSCLC (>70% tumor cells) and that the adjacent tissue specimen contained no tumor cells.

DNA was extracted from frozen lung tissue using the QIAamp DNA Kit (51306) and from blood samples using the QIAamp DNA Blood & Tissue Kit (69506). Total RNA was extracted from frozen lung tissue using the RNeasy Plus Kit (74134).

We performed WGS on matched tumor-blood samples from 92 cases and transcriptome sequencing on matched tumor-adjacent tissues from 90 of the same individuals. Clinical information and inferred TILs of 90 patients were listed in the Supplementary Table 7. An additional 57 cases (matched tumor-blood) were sequenced by WES.

**WGS and WES**. We performed 150 bp (WGS) and 100 bp (WES) paired-end sequencing reactions under contracts with WuXi NextCODE Co., Ltd. (Shanghai, China). The average sequence coverage was 67.84× for the tumor samples and 36.70× for the blood samples for WGS and 112.50× for the tumor samples and 111.80× for the blood samples for WES. The WES data were included to describe general mutation pattern of Chinese NSCLC patients and to identify the potential driver genes (Supplementary Fig. 1). The FastQC package (http://www.bioinformatics.babraham.ac.uk/projects/fastqc) was used to assess the quality-score distribution of the sequencing reads. Read sequences were mapped to the human reference genome (GRCh37) using the Burrows–Wheeler Aligner (BWA-MEM v0.7.15-r1140)[36] with the default parameters, and duplicates were marked and discarded using Picard (v1.70)(http://broadinstitute.github.io/picard). Then, the

local realignment and recalibration of the reads aligned by BWA was conducted using the Genome Analysis Toolkit (GATK v3.5)[37].

We included WGS data from 100 NSCLC patients (50 AD and 50 SCC) from TCGA project (Genomic Data Commons Data Portal: https://portal.gdc.cancer.gov/) and downloaded the aligned "bam" files for following analysis.

**Processing of genomic data.** Somatic substitutions and indels were detected using the Mutect2 mode in GATK on the GRCh37 genome build following the best practice for somatic SNV/indel calling (https://software.broadinstitute.org/gatk/best-practices/). Briefly, the algorithms compare the tumor to the matched normal sample to exclude germline variants. Somatic calls were excluded if: (1) they were found in a panel of normal built by 92 matched normal tissues; (2) they were located in the segmental duplication region marked by UCSC browser (http://genome.ucsc.edu/); (3) they were found in the 1000 genomes project (the Phase III integrated variant set release, across 2,504 samples) with the same mutation direction. Chromosome rearrangements and breakpoints were discovered using the local assembly tool novoBreak (v1.1.3rc)[38]. Breakpoints that were located in segmental duplication region marked by UCSC browser were excluded from analysis. Mutations (SNVs/indels) and breakpoints of chromosome rearrangements were annotated with the local versions of Oncotator (1.8.0)[39] and Annovar (2016Feb01)[40], according to GENCODE v19. BIC-Seq2 was used to address WGS data to detect somatic copy number alterations (CNAs) with the default parameters[41]. GATK best practices for somatic CNAs in exomes were used to detect CNAs from the WES data (https://software.broadinstitute.org/gatk/best-practices/).

**NSCLC drivers and comparison between the NJLCC and TCGA data.** The IntOGen platform[13] coupled with MutSigCV (1.4)[42] was used to identify SMGs. The IntOGen pipeline included two algorithms (OncodriveCLUST[43] and OncodriveFM[44]) that were designed to find genes with highly clustered mutations and non-randomly distributed functional mutations, respectively. MutSigCV was used to find genes with a higher mutation rate than the calculated background mutation rate. Multiple testing correction (Benjamini–Hochberg FDR) was performed separately and genes with $q$ values <0.05 in any algorithm were reported in this study (Supplementary Table 2). Then, we defined a SMG as a potential NSCLC driver gene if one of the following conditions was met: (1) the gene was collected by the COSMIC Cancer Gene Census as a mutated gene in lung cancer[12]; (2) the gene was reported as an IntOGen lung cancer gene[13]; or (3) the gene was a frequently altered gene reported in recent genomic studies on lung cancer[2]. Somatic mutations from TCGA WES data of 654 NSCLC patients (478 AD and 176 SCC) and clinical information were downloaded from the Firehose Broad GDAC (http://gdac.broadinstitute.org/, version 2016_01_28) to compare the mutation rates of the NSCLC genes.

**MS analysis.** MSs were identified using the Bioconductor package SomaticSignatures, which was based on the NMF methodology developed by Nik-Zainal et al.[9]. Briefly, we converted all mutation data from 92 WGS data sets into a matrix (M) composed of 96 features comprising mutation counts for each mutation type (C > A, C > G, C > T, T > A, T > C, and T > G) using each possible 5′ and 3′ context for all samples. MSs and their contribution to each sample's mutational spectrum were estimated with NMF decomposition method. We used the cosine similarity distance to measure the similarities between our identified signatures and the COSMIC signatures (http://cancer.sanger.ac.uk/cosmic/signatures). A MS-dominant patient was defined if the contribution of the MS was over 0.5 in the patient. The same analysis was applied to mutations called from 100 TCGA WGS data sets to further validate the identified MSs.

**Highly amplified/deleted regions and consensus clustering.** Copy number segments were used as input for GISTIC2[45] to identify significantly amplified/deleted regions with the default parameters. A default $q$ value threshold (0.25) was used to define highly amplified/deleted regions. To further identify subgroups of samples that shared similar CNA pattern, we performed consensus clustering using the ConsensusClusterPlus R package[46]. The input data for each sample were the copy number values for each identified region reported by GISTIC2. The copy numbers for each region were mean-centered across the samples prior to clustering. The following parameters were used in the consensus clustering: number of repetitions 1000; pItem = 0.7; pFeature = 0.7; Pearson distance metric and Ward linkage method. Different cluster solutions were evaluated.

**RSs and hotspot detection.** We extracted RSs from intra-chromosome rearrangements according to a previous study[11]. All intra-rearrangements were classified into deletions, duplications, and inversions, and then further sub-classified according to size of the rearranged segment (1–10 kb, 10–100 kb, 100 kb–1 Mb, 1–10 Mb, and more than 10 Mb). The classification produces a matrix of 15 distinct categories of structural variants across 92 NSCLC genomes. The matrix was decomposed using the same NMF methods applied for the extraction of MSs.

Three RSs were extracted from our lung cancer WGS data. A rearrangement was attributed to one RS if the probability of the rearrangement being generated from this specific signature was greater than 0.5 in a given sample. The probability of a given arrangement being assigned to one signature in the sample was

calculated as the probability of the RS generating this type of rearrangement, multiplied by the exposure of this sample to the signature.

A PCF algorithm was performed to define rearrangement hotspots with the default parameters ($\gamma = 8$, $k_{min} = 8$, and $i = 2$)[47]. Briefly, we collected both breakpoints of each rearrangement and sorted the breakpoints based on their chromosomal positions. Then, we calculated the inter-rearrangement distance (IMD) for each breakpoint, which was defined as the distance between one breakpoint and the breakpoint immediately preceding it. A putative hotspot was defined as a region having an average IMD that was two times greater than the genome-wide level and a breakpoint density higher than the expected number according to the background model.

Since numerous genomic features are known to affect the non-uniform distributions of rearrangements, we conducted a multi-variable genome-wide regression analysis according to a previous study[18] to evaluate the association between the genomic features and the rearrangements identified in our data to construct the background rearrangement model. Briefly, we divided the genome into non-overlapping genomic bins of 0.5 Mb and characterized each bin with features including the replication time, repetitive sequences, DNaseI hypersensitivity sites, non-mapping sites, known fragile sites and histone modification status of the chromatin. Then, to enable the comparability between different features, each feature was normalized to a mean of 0 and a standard deviation of 1 across the bin. We counted the total number of breakpoints for each bin and performed a negative binomial regression model to learn associated features. Finally, the properties obtained from the above model were used to calculate the expected number of breakpoints for the bin $f_i$ using the following formula:

$$b_i = e^m \prod_{i=1}^{N} e^{w_i f_i},$$

where $N$ represents the number of features, $w_i$ represents the weights of different features from the negative model, and $m$ represents the intercept of the model.

**RNA sequencing to assess gene expression and fusions.** Standard Illumina RNA-seq protocol (http://support.illumina.com/training/online-courses/sequencing.html) was conducted to generate transcription profiles of these samples using the Illumina HiSeq 1500 platform. RNA reads were generated, aligned to the GENCODE v19 genome assembly with STAR v2.4.1[48], and quantified with featureCounts v1.5.0[49]. The raw read counts were normalized with DESeq2[50] to estimate gene expression.

Fusions were detected by the somatic fusion genes finder FusionCatcher[51] with the default parameters, which integrated four commonly used aligners (Bowtie[52], BLAT[53], STAR[48], and Bowtie2[52]) to find reads of fusion transcripts. All fusions involving known functional fused genes reported in previous genomic studies of lung cancer[2,54] were included in the analysis. A new fusion involving BCAR4 was identified, because the count of unique reads mapping on the fusion junction was >10 and BCAR4 was recurrently fused in a TCGA NSCLC patient. A total of five important rearrangements were examined in the adjacent normal tissues by PCR, according to the breakpoints identified by both WGS and RNA-Seq in the matched tumor samples. The primers of PCR are listed in Supplementary Table 8.

Fusions in TCGA data were downloaded from ChimerDB v3.0[16]. All functional fused transcripts reported by previous studies were included in the analysis.

**GSEA analysis.** A GSEA was performed using Spearman's rank correlation coefficients between the MS3 proportion and the expression of genes using the "GSEA" command from the Bioconductor package clusterProfiler[55] based on 165 KEGG pathways (minimal set size: 20, maximal set size: 500) from the MSigDB (www.broadinstitute.org/gsea/msigdb) c2 database and T cell subpopulation-specific gene sets.

**Estimation of TILs.** We applied the tumor immune cell deconvolution method TIMER[56] to predict TILs by slightly adapting the available source code. TIMER is a recently developed computational approach that uses expression profiles from RNA-seq data to estimate the abundance of diverse TILs in tumor tissues. A curated leukocyte gene signature matrix was used as the reference data including 2271 signature genes overexpressed in the immune lineage. Normalized read counts were used as the input matrix for TIMER. Batch effects between our data and the external reference data sets were removed using ComBat command from the R package sva[57]. Six types of immune infiltrates were estimated (B cell, CD4+ T cell, CD8+ T cell, neutrophil, macrophage, and dendritic cell). TILs of TCGA patients were downloaded from the TIMER database (https://cistrome.shinyapps.io/timer/_w_7bef51bc/immuneEstimation.txt).

**T cell subpopulation-specific gene sets.** Raw RNA-seq data (Fastq) for the T lymphocyte subsets and B lymphocytes were downloaded from ArrayExpress (E-MTAB-2319)[24]. The same STAR-featureCounts-DESeq2 pipeline was used to quantify expression. Four types of T lymphocytes (T helper 1, T helper 2, T helper 17, and regulatory T cells) and B lymphocytes were included to identify cell-specific gene sets. The specificity measure (SPM) was used to evaluate the subpopulation lymphocyte subpopulation-specific expression patterns[58]. The SPM values range

from 0 to 1, with values close to 1 indicating a major contribution to gene expression in a specific lymphocyte relative to all other lymphocytes. For each T cell subpopulation, genes with SPMs higher than 0.9 were defined as subpopulation-specific genes.

**Immunohistochemistry analysis**. Immunohistochemistry method was used to validate the abundance of B cells and CD4+ T cells in 12 patients selected randomly from MS3-dominant patients defined above (Contribution$_{MS3}$ >0.5) and 12 patients selected randomly from other patients (Contribution$_{MS3}$ ≤0.5). Following the process of dewaxing and rehydratation, the sections were placed into a retrieval solution (G1202 pH 6.0) and then were heated to boiling for about 25 min. As cooled to room temperature, the sections were submerged in phosphate-buffered saline solution (pH 7.4) three times. Slides were further processed on an automated immunostainer. Endogenous peroxidase was blocked with 3% $H_2O_2$ solution shielded from light at room temperature. Next, slides were blocked with 3% BSA to avoid non-specific background staining. Afterward, the specimens were incubated overnight at 4 °C with a primary antibody (GB13064-1 Servicebio, for CD4, 1:100 dilution; ab134114, Abcam, for CD19, 1:300 dilution) and then were incubated for 1 h at room temperature with the biotinylated secondary antibodies diluted to 1:1000. The samples were developed using DAB-Plus Substrate Kit (invitrogen) under the microscope and were counterstained with hematoxylin, which was a nucleophilic dye. Finally, after dehydration and mounting, the specimens were observed with microscope and imaged for further analysis. To ensure the consistency of the results, the images were selected that were representative of different TILs levels, based on the results obtained from three pathologists without previously being acknowledged for the clinical information. In detail, the final results were present as the median number of TILs in the three hotspot areas selected randomly.

**Statistical analysis and figures**. All statistical tests were performed using a Wilcoxon rank-sum test for continuous data, Spearman's rank correlation or partial correlation for the estimation of correlation. Fisher's exact test was used to assess differences for the count data. Multiple testing corrections were performed where necessary using the Benjamini–Hochberg method. All reported $P$ values are two-sided. Figures were generated with the R packages ggplot2[59] and RColorBrewer[60].

**Data availability**. The sequencing data of this study have been deposited in the European Genome-phenome Archive (EGA) at the EMBL-European Bioinformatics Institute under accession number EGAD00001004071 (https://ega-archive.org/datasets/EGAD00001004071). The RNA-Seq data have been deposited at figshare (https://doi.org/10.6084/m9.figshare.6253973.v1). Data that support the findings of this study are available from TCGA database (http://cancergenome.nih.gov).

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

## Acknowledgements

This work was approved by the institutional review board of Nanjing Medical University and all the other participating institutions. This work was supported by the Science Fund for Creative Research Groups of the National Natural Science Foundation of China (81521004), National Key Research and Development Program of China (2017YFC0907905), Science Foundation for Distinguished Young Scholars in Jiangsu (BK20160046), National Natural Science Foundation of China (81573238, 81703295), "333 project" in Jiangsu Province, the Priority Academic Program for the Development of Jiangsu Higher Education Institutions (Public Health and Preventive Medicine), and the Top-notch Academic Programs Project of Jiangsu Higher Education Institutions (PPZY2015A067). The funders had no role in the study design, data collection and analysis, decision to publish, or preparation of the manuscript.

## Author contributions

H.S. and Z.H. initiated, conceived, and supervised the study. C.W. performed bioinformatics and statistical analysis with J.D., N.Q., M.Z., L.G., Y.W., and Q.W., and drafted the manuscript with Z.H. Y.G. performed the whole-genome sequencing/whole-exome sequencing and RNA-Seq experiments with T.J., Y.J., Z.P., F.D., J.H., and Y.Y., and conducted immunohistochemistry analysis with Z.Z. and J.Y. L.X., L.J., R.Y., S.C., J.H., L. C., L.D., and Y.Y. contributed pathology assessment and/or samples. Q.W., H.M., and G. J. proofed the manuscript.

## Additional information

**Competing interests:** The authors declare no competing interests.

