## [Peer Review File · Nature Communications]

Reviewers' comments:

Reviewer #1 (Remarks to the Author):

NCOMMS-17-19929-T

Wang et al. "Whole-genome sequencing reveals genomic signatures associated with the inflammatory microenvironments in Chinese NSCLC patients "

The authors perform genome wide mutation analyses on 149 Chinese non-small cell lung cancer (NSCLC) patient tumors including 92 whole genomes and 57 whole exome sequencing data. They also performed RNA-seq analyses on these tumors which were all from snap frozen tissues. Selected immunohistochemistry studies for tumor infiltrating lymphocytes (TILs) was performed along with computational biology analyses to determine the nature of the TILs in the tumor specimens from the RNA-seq data. They compare their results to deposited TCGA datasets for NSCLC. Their main findings are: the discovery of 27 significantly mutated genes (most of which have previously been described); the frequent occurrence of DNA loss in the HLA region, three types of mutational signatures (MS1, MS2-smoking, and MS3) of which the MS3 type was associated with the presence of EGFR mutations, large structural rearrangements, and the presence of large amounts of TILs. No analyses of clonal heterogeneity or calculation of neo-antigen load, or correlation with HLA region loss were presented. No correlation with any treatments administered were presented. No functional studies are presented.

Comments to the authors:

The paper is reviewed in the context of needing to learn more about both mutational spectrum and lung cancer pathogenesis, the relationship of lung cancer mutations to the tumor microenvironment, providing new information on the etiology of never smoking lung cancers, and the use of this information for directing patient care and follow-up.

The studies are technically well done. The size of the WGS sequencing information is comparable to that in TCGA. The information on TILs from the RNA seq data are interesting and potentially useful. The authors would do well in the abstract to state the frequency of the MS1, MS2, and MS3 mutational signature types and their relationship to adeno and squamous cell cancers. The following types of information would have greatly enhanced the value of this report.

1. Information on clinical treatment and follow-up related to the mutation status including targeted therapy, chemotherapy or immunotherapy. While these numbers are small they would be of value.
2. Information on whether any of the structural rearrangements they found could be detected in blood samples. The number they find is large. There are a few (like EML4-ALK and ROS) rearrangements which already guide targeted therapy treatment selection of NSCLCs. It would be extremely important and interesting to know if these rearrangements could be detected in "liquid biopsies" in the blood of these patients and thus could be used to follow the patients for disease status.
3. While the authors state a causal connection between the TILs and EGFR mutation, there is an association but not a causal connection demonstrated by their results and they need to edit for this.
4. Likewise if they could detect any of the rearrangements in histologically normal epithelium near the tumor this would be of great value because it could indicate the use of searching for such rearrangements for the early detection of lung cancer. Alternatively, it would be very important to know that such changes found in the tumor were not found in the surrounding histologically normal DNA (thus, either finding would increase the value of the manuscript).
5. Studies of "trunk" and "branch" structure (clonal heterogeneity) would have added value to the manuscript, although the authors may not be able to perform this given the depth of coverage they used for sequencing.
6. New clues to the etiology of never smoking lung cancer are important. While the authors describe

the mutational signature and mutations found it is not yet clear to me how this aids to our knowledge of the potential etiology or guides these studies forward. We know that immunologically "hot" tumors occur with a lot of mutations and do better with immunotherapy – is there a specific difference between Chinese and Caucasian patients in the etiology of never smoking lung cancer that their data has uncovered? If so, they need to clearly state this.

7. As part of this, I presume all of this data would be deposited so that others could "mine" it like the authors did with TCGA datasets.

Reviewer #2 (Remarks to the Author):

This is an important effort represent extensive whole genome and whole exome data sequencing from 149 non small cell lung cancer cases, coupled from TCGA analysis of an additional 100 cases. The primary finding is of 3 mutational signatures (MS1, MS2 and MS3), with MS1 being correlated with APOPEC, MS2 with smoking history, and MS3 with inflammatory tumour infiltrating B lymphocytes, and with EGFR mutations. This signature was also correlated with a higher level of chromosomal rearrangements.

An important question remains to what extent the mutation signatures identified in this population differ to those identified previously. In particular:

Is MS1 merely a combination of the two Cosmic signatures 'signature 2' and 'signature 13'. Previous studies of lung cancer have also tended to show an association with 'signature 2' and not 'signature 13'. It is again curious why, in this data, the stronger association appears to be with 13. Could this be due to the relatively limited number of cases.

The greater frequency of MS1 among non-smoking female cases compared to non-smoking LUAD male cases is also unclear.

The important contribution of the S2 signature (ie related to smoking) that was present in all LUAD cases, and predominantly so for 2 cases needs comment. Similarly, the lack of any difference between smoking and non-smoking LUSC cases for MS2 is also curious. Either these cases are actually smokers, or these results suggest an alternative exposure to tobacco related carcinogens.

Plans for deposition of the genomic data in public databases subsequent to any publication need to be clarified

Reviewer #3 (Remarks to the Author):

In this manuscript, Wang C et al take advantage of WGS (92 cases), WES (57 cases) and RNA-Seq (90 cases) of Chinese NSCLC specimens to functionally characterize the genomic signature and inflammatory infiltrating lymphocyte pattern which I think is comprehensive and significant. The authors revealed the largely different genetic mutational pattern in Chinese NSCLC in comparison with the TCGA data mainly based on Caucasian population. Interestingly, they demonstrated that Chinese NSCLC patients are more likely to fit the type3 mutational signature (MS3) with a correlation of Type 1/2 rearrangement signatures (RS1/2). Moreover, they developed algorithms to uncover the association between the MS3 and tumor infiltrating lymphocytes (mainly CD19+ B cells and CD4+ T cells) as well as the EGFR mutation rates. These findings are of great interest. However, a few questions/comments should be addressed before the publication.

I list my concerns below:

- 1) The finding of correlation between TIL and EGFR mutation is interesting. However, without any experimental evidence, I am not convinced that "inflammatory infiltration may contribute to the accumulation of EGFR mutations". Instead, it remains possible that EGFR mutation triggers inflammatory infiltration of B lymphocytes into tumors.
- 2) Patient consent and ethical-related information are missing throughout the manuscript. It's also not clear to me if any of these patients received any treatment or neoadjuvant therapy before surgery/biopsy. If yes, how the MS3 correlates with disease status or therapeutic response. Moreover, it'll be interesting to test the MS3 dynamics using biopsy samples before and after TKI-treatment.
- 3) Lung cancer is highly heterogeneous and this also applies to infiltrating lymphocytes. Please provide a table with detailed counting of various lymphocytes in all lung cancer samples.
- 4) In Fig 3d, the authors claim 10 hotspots in the text. It seems only 9 in the figure.

Reviewer #4 (Remarks to the Author):

Title: Whole-genome sequencing reveals genomic signatures associated with the inflammatory microenvironments in Chinese NSCLC patients

Summary:

The authors determined the genomic signatures based on mutation and copy number alternation (CNA) profiles in Chinese patients with NSCLC including lung ADC and SCC patients by evaluating whole-genome sequencing (WGS) (n=96). RNA-Seq was also utilized to assess gene expression of tumor-adjacent tissues for 90 patients having genome profiled by WGS. From the mutation profiles, the authors found that the patients (1) had EGFR frequently mutated (the lung ADC cases were almost entirely non-smoking women), and (2) were characterized by three mutational signatures (MS) corresponding to APOBEC- (referred to as MS1), smoking- (MS2) and unknown, predominantly observed in the cohort MS3. They also found the MS3 signature was associated with either EGFR mutation or fusion involved genes relating to the RTK pathway. From the CNA profiles, they identified three rearrangement signatures (RS) based on characterizing size and types (i.e. deletion, duplication and inversion) of CNAs. The RS1 and 2 characterized by large size CNAs were associated with MS3, while the RS3, which was known for association with BRCA2 mutation in breast cancer, was related to smoking MS2 signature. Finally, by integrating with gene expression of tumor-adjacent tissues the authors found the association between MS3 and in silico estimated tumor-infiltrating lymphocytes (TILs), especially B and CD4+ T cells. GSEA also revealed that the highly expressed genes in patients carrying MS3 signature were enriched for Treg, Th2 and Th17 gene signatures. The association between MS3 and TILs was validated by IHC for 12 MS3 high and 12 MS3 low patients.

Comments:

1. The manuscript has flaws in construction and writing. For instance, figures have typos (Fig. 3c), missing axis titles (Fig. 1a) or are incorrectly labeled (Ext. Fig. 6a). Figure title (e.g. Fig 2) incorrectly described the figure content.
2. The authors initially described two groups of patients were subjected for genomic study: WGS (n=96) and WES (n=57). However, most figures were derived from 96 WGS. What results were obtained from WES?
3. The MS3 group was similar to Signature 1B (Alexandrov 2013 or citation # 7) which is related to age. The authors should re-examine the correlation of MS3 and signature 1B, as well as the association of MS and patient age in the cohort. If MS3 is actually correlated with age, then the association of MS3 with TIL could be due to an age-related pattern, instead of a causal relationship implicating MS3 in high level TILs. Furthermore, the authors should determine if the mutation pattern in EGFR gene carries an MS3 signature (C>T pattern).

Reviewers' comments:

Reviewer #1 (Remarks to the Author):

NCOMMS-17-19929-T

Wang et al. "Whole-genome sequencing reveals genomic signatures associated with the inflammatory microenvironments in Chinese NSCLC patients "

The authors perform genome wide mutation analyses on 149 Chinese non-small cell lung cancer (NSCLC) patient tumors including 92 whole genomes and 57 whole exome sequencing data. They also performed RNA-seq analyses on these tumors which were all from snap frozen tissues. Selected immunohistochemistry studies for tumor infiltrating lymphocytes (TILs) was performed along with computational biology analyses to determine the nature of the TILs in the tumor specimens from the RNA-seq data. They compare their results to deposited TCGA datasets for NSCLC. Their main findings are: the discovery of 27 significantly mutated genes (most of which have previously been described); the frequent occurrence of DNA loss in the HLA region, three types of mutational signatures (MS1, MS2-smoking, and MS3) of which the MS3 type was associated with the presence of EGFR mutations, large structural rearrangements, and the presence of large amounts of TILs. No analyses of clonal heterogeneity or calculation of neo-antigen load, or correlation with HLA region loss were presented. No correlation with any treatments administered were presented. No functional studies are presented.

Comments to the authors:

The paper is reviewed in the context of needing to learn more about both mutational spectrum and lung cancer pathogenesis, the relationship of lung cancer mutations to the tumor microenvironment, providing new information on the etiology of never smoking lung cancers, and the use of this information for directing patient care and follow-up.

The studies are technically well done. The size of the WGS sequencing information is comparable to that in TCGA. The information on TILs from the RNA seq data are interesting and potentially useful. The authors would do well in the abstract to state the frequency of the MS1, MS2, and MS3 mutational signature types and their relationship to adeno and squamous cell cancers. The following types of information would have greatly enhanced the value of this report.

Response: We thank the reviewer for the encouraging comments and suggestions that help enhance the value of our study. In the abstract of the revised version, we added a short description as below:

"To reveal the underlying mechanisms, we present a comprehensive genomic landscape of 149 Non-Small Cell Lung Cancer (NSCLC) cases. We identify 15 potential driver genes and three mutational signatures, two signatures have been connected with APOBEC enzymes (MS1, 6.5%) and smoking (MS2, 47.8%). In addition, we reveal that Chinese patients are specially characterized by not only highly clustered and functional *EGFR* mutations but also

a mutational signature (MS3, 33.7%) that is associated with inflammatory tumor-infiltrating B lymphocytes ($P = 0.001$). The *EGFR* mutation rate is significantly increased with the proportion of the MS3 signature ($P=9.37\times 10^{-5}$).”

1. Information on clinical treatment and follow-up related to the mutation status including targeted therapy, chemotherapy or immunotherapy. While these numbers are small they would be of value.

Response: In response to the reviewer’s comment, we provide clinical treatment information of 90 patients with both WGS and RNA-Seq data in Supplementary Table 7. More than a half of the patients (48 cases, 53.3%) were treated by chemotherapy or chemoradiotherapy after surgical resection. Only a few patients (5 cases) received the targeted therapy of TKI inhibitors and none of patients received immunotherapy. We did not observe any significant association between clinical treatment and mutation statuses (mutations in driver genes and mutational signatures). Because the majority of the tumor involved in this study (97.8%) were Stage I-III lung cancer and the follow-up was not long enough, the case fatality rate is relatively low (17.8%). Thus, we did not use the follow-up information in this study.

2. Information on whether any of the structural rearrangements they found could be detected in blood samples. The number they find is large. There are a few (like EML4-ALK and ROS) rearrangements which already guide targeted therapy treatment selection of NSCLCs. It would be extremely important and interesting to know if these rearrangements could be detected in “liquid biopsies” in the blood of these patients and thus could be used to follow the patients for disease status.

Response: We agree with the reviewer about the potential value of rearrangements in “liquid biopsies”. As far as we know, however, plenty of peripheral blood and special manipulation after sample collection is required to ensure detection sensitivity and accuracy of genomic alterations in blood^{1,2}. Because the initial aim of this study was to describe a landscape of genomic alterations, the blood samples of the patients in this study were collected as the germline control of tumor tissues and were not suitable for the rearrangement detection after careful evaluation. Nevertheless, we are very interested in the detection of “liquid biopsies” and have proposed a new study proposal focusing on the cell free DNA (cfDNA) based on the qualified samples.

3. While the authors state a causal connection between the TILs and *EGFR* mutation, there is an association but not a causal connection demonstrated by their results and they need to edit for this.

Response: We agree with the reviewer that the causal connection between tumor infiltrating lymphocytes (TILs) and *EGFR* mutations needs further evaluation through experiments. In revision, we presented additional evidence. We evaluated the evolution of microenvironments using “trios” samples from a recently published study³, which included tumor tissues, precancerous tissues (atypical adenomatous hyperplasia, AAH) and normal tissues. We inferred the abundance of TILs by using RNA-Seq data (GSE102511) and found that the inflammatory TILs (B cells and CD4+ T cells) increased not only in tumor biopsies but also in premalignant

tissues (AAH), but the majority of *EGFR* mutations occurred only in tumor tissues (Response Figure 1). These results suggested that the inflammatory environments formed ahead of the genomic alterations (e.g., *EGFR* mutations) and thus provided evidence for the potential causal association between inflammatory TILs and *EGFR* mutations. However, further experiments are warranted to illuminate the causal connection and the underlying mechanisms. Thus, we edited the discussion in the revised version (Page 12-13):

“Our results further identified the association between inflammatory TILs and well-known *EGFR* mutations in Chinese NSCLC patients. In addition, we reanalyzed sequencing data from a recent study on lung adenocarcinoma and precancerous tissues. We found that inflammatory microenvironments formed earlier than the occurrence of *EGFR* mutations (Supplementary Fig. 10). Because inflammatory microenvironments release signaling molecules such as the tumor growth factor EGF, which can serve as a ligand for the EGFR protein, we proposed that the microenvironments may select tumor cells with an EGFR protein that is highly activated by functional mutations in *EGFR*, which provides a new potential explanation for the frequent and recurrent *EGFR* mutations in never-smokers. Thus, patients with *EGFR* mutations may benefit from a combination of anti-tumor immunologic therapy and an EGFR inhibitor, especially TKI-resistant individuals, although the causal relationship between TILs and *EGFR* mutations needs further evaluation.”

Response Figure 1. Abundance of B cells and T cells (CD4+) in normal, AAH and Lung AD. Samples with *EGFR* mutations were colored as red.

4. Likewise if they could detect any of the rearrangements in histologically normal epithelium near the tumor this would be of great value because it could indicate the use of searching for such rearrangements for the early detection of lung cancer. Alternatively, it would be very important to know that such changes found in the tumor were not found in the surrounding histologically normal DNA (thus, either finding would increase the value of the manuscript).

Response: We thank the reviewer for the helpful suggestions. Among the patients with important rearrangements mentioned in Figure 2, only 5 patients have enough adjacent normal tissues for detection. Thus, we examined the 5 important rearrangements (Figure 2) in the adjacent normal

tissues by PCR, according to the breakpoints identified by both WGS and RNA-Seq in the matched tumor samples. We did not identify any rearrangement in the adjacent normal tissues of these patients (Response Figure 2). The primer of PCR were listed in Response Table 1. We mentioned the results in Page 8.

Response Figure 2. Electrophoretogram of rearrangements PCR products in adjacent normal tissues

Response Table 2. The primer for the detection of rearrangements in adjacent normal tissues

Sample	Fused gene	Primer 1	Primer 2
NJLCC_0067	EML4-ALK	Left: TTCTTAGCAGCAACAGGTG	Left: GCCAGTCCTCAATAATTCA
		Right: CTCCTGCCCTGTTCCCTA	Right: GGACTGCAGTTTCCCTCTCT
NJLCC_0031	KIF5B-RET	Left: ACATTTCAAAATTGGCTCC	Left: AGTCTTCAGTGCCTTAGTTT
		Right: CTTCCCTGCCGCTGTCAC	Right: ACAGTCAAGGTCAGTGTCG
NJLCC_0066	EZR-ROS1	Left: CGCGAGAAGGAGGAGTTG	Left: CGTGGAGAGAGAGAAAGA
		Right: AT	Right: GCA

		CCAAAGGTCAGTGGGATT GT	CAAAGGTCAGTGGGATTGT AACA
NJLCC_00 59	CD74-ROS 1	Left: AGGCTGGTCTTGAACCTCT G Right: TTAGCATGCCAAGACCAA CG	Left: CCACCACGCCCAGCTAAT Right: TGATGCATGTGAGTCCTTT AACT
NJLCC_00 48	CD63-BCA R4	Left: ATCATGTTGGTGGAGGTGG C Right: GGTCTCGAACTCCTCACCT C	Left: GGGCCCTGTAATGCATAG A Right: ACAGACATAAGCACCACT TG

5. Studies of “trunk” and “branch” structure (clonal heterogeneity) would have added value to the manuscript, although the authors may not be able to perform this given the depth of coverage they used for sequencing.

Response: We agree with the reviewer’s comments. However, an evident evolution analysis to study “trunk” and “branch” structure commonly requires: (1) multi-region biopsies from a single patients (2) ultra-deep depth of sequencing, as indicated by reviewer. Our current study did not include multi-region samples, thus it is difficult to distinguish “trunk” and “branch” mutations. In the revision, we conducted subclone analysis in a sample with the highest coverage. We estimated allelic-specific copy number alterations by *ascatNGS*⁴ and inferred the clonal population structure by *PyClone*⁵. However, the subclones cannot be properly estimated because of the great variance of variant allele frequency (VAF) caused by limited depth. Thus, we did not include the analysis in this study.

6. New clues to the etiology of never smoking lung cancer are important. While the authors describe the mutational signature and mutations found it is not yet clear to me how this aids to our knowledge of the potential etiology or guides these studies forward. We know that immunologically “hot” tumors occur with a lot of mutations and do better with immunotherapy – is there a specific difference between Chinese and Caucasian patients in the etiology of never smoking lung cancer that their data has uncovered? If so, they need to clearly state this.

Response: We thank the reviewer’s suggestion. Limited by the sample size, our genomic study in 149 NSCLC patients in China cannot tell if there is any specific difference between Chinese and Caucasian patients in the etiology of never smoking lung cancer; however, our results revealed that *EGFR* mutation rate was much higher in our never-smoking lung AD patients (62%) than in TCGA never-smoking AD patients (37%), partially suggesting that the etiology of lung cancer in never-smokers between Chinese and Caucasian patients should be different.

Unlike the anti-tumor microenvironments (e.g., infiltrating CD8+ cells) involved in the immunotherapy and high mutation burden in Caucasian patients, our results suggested that tumor-prompting inflammatory microenvironments (B cells and tumor-stimulating regulatory T cells) could be one of the underlying mechanisms. The similar pattern was also found in a recent study on the early stage lung adenocarcinoma⁶, emphasizing the potential role of the inflammatory microenvironment in the initiation of lung cancer of never-smokers. Recent studies on Chinese lung cancer patients also provided evidence for the association between chronic inflammation and non-smoking lung cancer: Shiels et al. found that circulating inflammation markers were elevated in the lung cancer patients several years before the diagnosis⁷; Our recent work also suggested that circulating polyunsaturated fatty acids, which are important participants in the stimulation of chronic inflammation, were causal risk factors of lung cancer, especially for female never-smokers⁸.

Moreover, our results further connected the microenvironments with genomic features. In this revision, we provided additional evidence that the increase of inflammatory microenvironments formed before *EGFR* mutations, which provided new clues for the causal association. Thus, we proposed a “microenvironments-driven” model for the never-smokers in Chinese. With these evidence, additional cell and animal models are warranted to investigate the evolution of mutations under certain microenvironments and illuminate the underlying mechanisms. We revised the discussion in Page 12.

7. As part of this, I presume all of this data would be deposited so that others could “mine” it like the authors did with TCGA datasets.

Response: We will deposit the data in the Genome Sequence Archive of Beijing Institute of Genomics (<http://gsa.big.ac.cn/>)⁵, which is a data repository in compliance with data standards and structures of the International Nucleotide Sequence Database Collaboration (INSDC).

Reviewer #2 (Remarks to the Author):

This is an important effort represent extensive whole genome and whole exome data sequencing from 149 non small cell lung cancer cases, coupled from TCGA analysis of an additional 100 cases. The primary finding is of 3 mutational signatures (MS1, MS2 and MS3), with MS1 being correlated with APOBEC, MS2 with smoking history, and MS3 with inflammatory tumour infiltrating B lymphocytes, and with EGFR mutations. This signature was also correlated with a higher level of chromosomal rearrangements.

An important question remains to what extent the mutation signatures identified in this population differ to those identified previously. In particular:

Is MS1 merely a combination of the two Cosmic signatures 'signature 2' and 'signature 13'. Previous studies of lung cancer have also tended to show an association with 'signature 2' and not 'signature 13'. It is again curious why, in this data, the stronger association appears to be with 13. Could this be due to the relatively limited number of cases.

Response: We understand the reviewer's concerns. Yes, MS1 in our study should be a combination of the two known signatures. We found that the cosine similarity between MS1 and COSMIC signature 2 and 13 were 0.82 and 0.84 respectively, and the cosine similarity between MS1 and sum of the two signatures were 0.99. According to the description of signature 2 and 13 on the COSMIC website, "the Signature 2 is usually found in the same samples as Signature 13" (<http://cancer.sanger.ac.uk/cosmic/signatures>). In a recent large genomic study of lung cancer⁹, both signatures were also identified. In addition, both signatures have been attributed to the aberration of APOBEC enzymes, thus we used the combined signature MS1 to describe the general pattern of both signatures and mainly focused on MS3 signature in this study.

The greater frequency of MS1 among non-smoking female cases compared to non-smoking LUAD male cases is also unclear.

Response: We evaluated the association between MS1 and gender in non-smoking LUAD patients. The proportion of MS1 level was higher in female cases (n=23; 67.6%) than that in males (n=11; 32.3%)($P_{wilcoxon}=0.07$). Because MS1 was an APOBEC-related signature, we investigated the expression pattern of major APOBEC enzymes (APOBEC1, APOBEC3A/B) reported by a previous study¹⁰. We found that the expression of APOBEC3A was significantly higher in females than in the males with a similar association ($P_{wilcoxon}=0.07$, Response Figure 3), suggesting that a relatively higher activation rate of APOBEC3A in female cases may be responsible for the greater MS1 signature in non-smoking females. However, because of the limited sample size in the stratification analysis, we did not describe and discuss the results in this study.

Response Figure 3. Association between the expression of APOBEC1/APOBEC3A/APOBEC3B and gender.

The important contribution of the S2 signature (ie related to smoking) that was present in all LUAD cases, and predominantly so for 2 cases needs comment. Similarly, the lack of any difference between smoking and non-smoking LUSC cases for MS2 is also curious. Either these cases are actually smokers, or these results suggest an alternative exposure to tobacco related carcinogens.

Response: The reviewer raises an important question about the inconsistency between smoking related MS2 and smoking history in specific patients. In our study, each patient was interviewed face-to-face by well-trained investigators with epidemiology background and the basic smoking exposure information should be correct. Consistent with previous studies of lung cancer⁹, the MS2 signature (COSMIC signature 4) was commonly seen in the never-smokers, although the proportion was relatively low.

Moreover, in order to carefully check the status of two never-smoking lung AD cases with high MS2 signature, we conducted additional follow-up and one female patient responded to our interview. The patient is a life-long non-smoker and there is no smoker in her family members living together or colleagues working together, but she was exposed to cooking fume without any protection. Exposure to cooking fume was one of the risk factors of lung cancer in Chinese female non-smokers¹¹. However, we cannot draw any conclusion from a single patient and additional studies are warranted to further illuminate the connection between cooking fume and the smoking signature.

In addition, previous researchers⁹ tried to use the signature to classify all patients into smokers and non-smokers and achieved a more consistent result in AD (area under the curve (AUC) =0.87) than in SCC (AUC=0.62), suggesting that the discrepancy occurred more frequently in SCC. Moreover, only 6 never-smoking SCC patients were included in this analysis, which may result in the lack of difference between smoking and non-smoking SCC cases for MS2.

Plans for deposition of the genomic data in public databases subsequent to any publication need to

be clarified

Response: We will deposit the data in the Genome Sequence Archive of Beijing Institute of Genomics (<http://gsa.big.ac.cn/>)⁵, which is a data repository in compliance with data standards and structures of the International Nucleotide Sequence Database Collaboration (INSDC).

Reviewer #3 (Remarks to the Author):

In this manuscript, Wang C et al take advantage of WGS (92 cases), WES (57 cases) and RNA-Seq (90 cases) of Chinese NSCLC specimens to functionally characterize the genomic signature and inflammatory infiltrating lymphocyte pattern which I think is comprehensive and significant. The authors revealed the largely different genetic mutational pattern in Chinese NSCLC in comparison with the TCGA data mainly based on Caucasian population. Interestingly, they demonstrated that Chinese NSCLC patients are more likely to fit the type3 mutational signature (MS3) with a correlation of Type 1/2 rearrangement signatures (RS1/2). Moreover, they developed algorithms to uncover the association between the MS3 and tumor infiltrating lymphocytes (mainly CD19+ B cells and CD4+ T cells) as well as the EGFR mutation rates. These findings are of great interest. However, a few questions/comments should be addressed before the publication.

Response: We thank the reviewer's positive comments.

I list my concerns below:

1) The finding of correlation between TIL and EGFR mutation is interesting. However, without any experimental evidence, I am not convinced that “inflammatory infiltration may contribute to the accumulation of EGFR mutations”. Instead, it remains possible that EGFR mutation triggers inflammatory infiltration of B lymphocytes into tumors.

Response: We agree with the reviewer that the causal connection between tumor infiltrating lymphocytes (TILs) and *EGFR* mutations needs further evaluation through experiments. In revision, we presented additional evidence. We evaluated the evolution of microenvironments using “trios” design, which included tumor tissues, precancerous tissues (atypical adenomatous hyperplasia, AAH) and normal tissues. We inferred the abundance of TILs by using RNA-Seq data downloaded from a previously published study³ (GSE102511) and found that the inflammatory TILs (B cells and CD4+ T cells) increased not only in the tumor biopsies but also premalignant tissues, but the majority of *EGFR* mutations occurred only in tumor tissues (Response Figure 4). The results suggested that the inflammatory environments formed ahead of the genomic alterations (e.g., *EGFR* mutations) and provided evidence for the causal association between inflammatory TILs and *EGFR* mutations. However, further experiments are warranted to further illuminate the causal connection and the underlying mechanisms. Thus, we edited the discussion in the revised version (Page 12-13):

“Our results further identified the association between inflammatory TILs and well-known *EGFR* mutations in Chinese NSCLC patients. In addition, we reanalyzed sequencing data from a recent study on lung adenocarcinoma and precancerous tissues. We found that inflammatory microenvironments formed earlier than the occurrence of *EGFR* mutations (Supplementary Fig. 10). Because inflammatory microenvironments release signaling molecules such as the tumor growth factor EGF, which can serve as a ligand for the EGFR protein, we proposed that the microenvironments may select tumor cells with an EGFR protein that is highly activated by functional mutations in *EGFR*, which provides a new potential explanation for the frequent and recurrent *EGFR* mutations in never-smokers.

Thus, patients with *EGFR* mutations may benefit from a combination of anti-tumor immunologic therapy and an *EGFR* inhibitor, especially TKI-resistant individuals, although the causal relationship between TILs and *EGFR* mutations needs further evaluation.”

Response Figure 4. Abundance of B cells and T cells (CD4+) in normal, AAH and Lung AD. Samples with *EGFR* mutations were colored as red.

2) Patient consent and ethical-related information are missing throughout the manuscript. It’s also not clear to me if any of these patients received any treatment or neoadjuvant therapy before surgery/biopsy. If yes, how the MS3 correlates with disease status or therapeutic response. Moreover, it’ll be interesting to test the MS3 dynamics using biopsy samples before and after TKI-treatment.

Response: We understand the reviewer’s concern. In the current study, we only included patients who did not receive any treatment or neoadjuvant therapy before surgery/biopsy. We added the description in the revised methods (Page 14).

This study and its design were approved by the local ethics committee (Nanjing Medical University and Shanghai Chest Hospital) and all participants provided written informed consent for the research. We have provided the information in the revised methods (Page 14).

3) Lung cancer is highly heterogeneous and this also applies to infiltrating lymphocytes. Please provide a table with detailed counting of various lymphocytes in all lung cancer samples.

Response: As suggested by the reviewer, we have added the detailed counting of six types of TILs in Supplementary Table 7.

4) In Fig 3d, the authors claim 10 hotspots in the text. It seems only 9 in the figure.

Response: We are sorry for the unclear description. In total, we identified 22 hotspots. Two of them at 2p22.3 (*LINC00486*) and 2p25.3 (*TPO-PXDN*) showed an extremely high rearrangement rate (>100/Mb). Thus, we did not present them in Figure 3d. Among the rest of hotspots, 10 overlapped with copy number altered regions identified in this study (green triangle), 9 of them

overlapped with reported fragile sites (red square) and 9 of them overlapped with copy number altered regions identified in the TCGA NSCLC data (blue diamond). We have modified the description to make it clear and added information in the legend of Figure 3d (Page 9).

Reviewer #4 (Remarks to the Author):

Title: Whole-genome sequencing reveals genomic signatures associated with the inflammatory microenvironments in Chinese NSCLC patients

Summary:

The authors determined the genomic signatures based on mutation and copy number alternation (CNA) profiles in Chinese patients with NSCLC including lung ADC and SCC patients by evaluating whole-genome sequencing (WGS) (n=96). RNA-Seq was also utilized to assess gene expression of tumor-adjacent tissues for 90 patients having genome profiled by WGS. From the mutation profiles, the authors found that the patients (1) had EGFR frequently mutated (the lung ADC cases were almost entirely non-smoking women), and (2) were characterized by three mutational signatures (MS) corresponding to APOBEC- (referred to as MS1), smoking- (MS2) and unknown, predominantly observed in the cohort MS3. They also found the MS3 signature was associated with either EGFR mutation or fusion involved genes relating to the RTK pathway. From the CNA profiles, they identified three rearrangement signatures (RS) based on characterizing size and types (i.e. deletion, duplication and inversion) of CNAs. The RS1 and 2 characterized by large size CNAs were associated with MS3, while the RS3, which was known for association with BRCA2 mutation in breast cancer, was related to smoking MS2 signature. Finally, by integrating with gene expression of tumor-adjacent tissues the authors found the association between MS3 and in silico estimated tumor-infiltrating lymphocytes (TILs), especially B and CD4+ T cells. GSEA also revealed that the highly expressed genes in patients carrying MS3 signature were enriched for Treg, Th2 and Th17 gene signatures. The association between MS3 and TILs was validated by IHC for 12 MS3 high and 12 MS3 low patients.

Comments:

1. The manuscript has flaws in construction and writing. For instance, figures have typos (Fig. 3c), missing axis titles (Fig. 1a) or are incorrectly labeled (Ext. Fig. 6a). Figure title (e.g. Fig 2) incorrectly described the figure content.

Response: We apologize for the flaws in the manuscript. In the revision, we have carefully checked the whole manuscript and verified the figures as well as related description. We have corrected the typos in Figure 3c, added axis titles for Figure 1a and corrected the error figure in Supplementary Fig. 7a (original Ext. Fig. 6a). The title of Figure 2 was also corrected in the article.

2. The authors initially described two groups of patients were subjected for genomic study: WGS (n=96) and WES (n=57). However, most figures were derived from 96 WGS. What results were obtained from WES?

Response: In this study, we only included the WES data in the Figure 1 and related Supplementary Fig. 2 to describe the general mutation pattern of Chinese NSCLC patients and to identify the potential significantly mutated genes, which was one of the important sections of this study. The general design of this study was presented in Supplementary Fig. 1. We added more description about the connection between the data and our results in the revised methods (Page

14).

3. The MS3 group was similar to Signature 1B (Alexandrov 2013 or citation # 7) which is related to age. The authors should re-examine the correlation of MS3 and signature 1B, as well as the association of MS3 and patient age in the cohort. If MS3 is actually correlated with age, then the association of MS3 with TIL could be due to an age-related pattern, instead of a causal relationship implicating MS3 in high level TILs. Furthermore, the authors should determine if the mutation pattern in EGFR gene carries an MS3 signature (C>T pattern).

Response: In response to the reviewer's comment, we re-examined the correlation of MS3 and Signature 1B. We observed that the two signatures were similar (Cosine similarity=0.92). However, the Signature 1B was believed to be less efficient separation from other signatures in some cancer types because of the limited sample size according to the statement of Alexandrov's work¹⁰. Thus, we also evaluated the similarity between Signature 1A, Signature 1B and Signature 5 reported by Alexandrov's study. We found that Signature 1B was highly similar with the combined signature of Signature 1A and Signature 5 (Cosine similarity=0.97). Thus, we thought that Signature 1B of Alexandrov was a combination of Signature 1A and Signature 5. In addition, we did not observe a significant association between age and MS3 ($P_{\text{Spearman}} = 0.48$, $r_{\text{Spearman}} = -0.08$) in our data. For *EGFR* mutations, we cannot distinguish the mutational signature of *EGFR* because the majority of *EGFR* substitutions mutated from T to G at 858th amino acid (L858R) and the context nearby were CTG and the mutations could be classified as any mutational signature identified in this study.

Reference:

1. Wan JCM, *et al.* Liquid biopsies come of age: towards implementation of circulating tumour DNA. *Nature reviews Cancer* **17**, 223-238 (2017).
2. Heitzer E, Perakis S, Geigl JB, Speicher MR. The potential of liquid biopsies for the early detection of cancer. *npj Precision Oncology* **1**, 36 (2017).
3. Sivakumar S, *et al.* Genomic Landscape of Atypical Adenomatous Hyperplasia Reveals Divergent Modes to Lung Adenocarcinoma. *Cancer research* **77**, 6119-6130 (2017).
4. Raine KM, *et al.* ascatNgs: Identifying Somatically Acquired Copy-Number Alterations from Whole-Genome Sequencing Data. *Current protocols in bioinformatics* **56**, 15 19 11-15 19 17 (2016).
5. Roth A, *et al.* PyClone: statistical inference of clonal population structure in cancer. *Nature methods* **11**, 396-398 (2014).
6. Lavin Y, *et al.* Innate Immune Landscape in Early Lung Adenocarcinoma by Paired Single-Cell Analyses. *Cell* **169**, 750-765 e717 (2017).

7. Shiels MS, *et al.* A prospective study of immune and inflammation markers and risk of lung cancer among female never smokers in Shanghai. *Carcinogenesis* **38**, 1004-1010 (2017).
8. Wang C, *et al.* Metabolome-wide association study identified the association between a circulating polyunsaturated fatty acids variant rs174548 and lung cancer. *Carcinogenesis* **38**, 1147-1154 (2017).
9. Campbell JD, *et al.* Distinct patterns of somatic genome alterations in lung adenocarcinomas and squamous cell carcinomas. *Nature genetics* **48**, 607-616 (2016).
10. Alexandrov LB, *et al.* Signatures of mutational processes in human cancer. *Nature* **500**, 415-421 (2013).
11. Zou XN, *et al.* Histological subtypes of lung cancer in Chinese women from 2000 to 2012. *Thoracic cancer* **5**, 447-454 (2014).

REVIEWERS' COMMENTS:

Reviewer #1 (Remarks to the Author):

The authors have responded appropriately to the reviewers' comments including providing additional new data.

Reviewer #2 (Remarks to the Author):

No additional comments

Reviewer #3 (Remarks to the Author):

The manuscript has been significantly improved and I have no more concerns.

Reviewer #4 (Remarks to the Author):

Wang et al.

Whole-genome sequencing reveals genomic signatures associated with the inflammatory microenvironments in Chinese NSCLC patients

1. The authors have addressed the comment regarding typos and writing.
2. The authors have also clarified when WES and WGS data have been used. The manuscript has been modified to address this point.
3. The review has raised questions regarding the similarity between MS3 and Signature 1B originally reported in Alexandrov et al. (citation #7). The authors have found that even though Signature 1B is similar to MS3 (cosine similarity = 0.92), the original Signature 1B is actually the combination of the other original Signature 1A and Signature 5. According to Alexandrov et al. (2013), "Signatures 1A/B, 2 and 5 were also found in lung adenocarcinoma. Signature 5, but not Signatures 1A/B and 2, also showed a positive correlation between smoking history and mutation contribution ($P = 8.0 \times 10^{-3}$, Supplementary Fig. 96). Thus, in lung cancer, Signature 5, which is characterized predominantly by C>T and T>C mutations, may also be due to tobacco carcinogens." Because MS3 is found in female never-smokers in this study, it may be related to secondary smokers, while MS2 is from primary smokers.
4. The causal relationship between EGFR mutation and TIL (Reviewer 1's Comment 3 and Reviewer 3's Comment 1).

In response to the critiques of Reviewers 1 and 3, the authors demonstrate that the inflammatory microenvironment may occur before occurrence of EGFR mutations (Response Figure 1) using data derived from a recent publication. To more convincingly show that the inflammatory microenvironment preceded EGFR mutation, the authors could also show data demonstrating that the level of infiltrating lymphocytes in AAH are higher in individuals with EGFR mutant ADC as compared to AAH lesions in tumors that do not have EGFR mutation. Inferring temporal relationships and causality from a single point in time remains problematic, as the authors imply.

REVIEWERS' COMMENTS:

Reviewer #1 (Remarks to the Author):

The authors have responded appropriately to the reviewers' comments including providing additional new data.

Response: We thank the reviewer for the positive comment.

Reviewer #2 (Remarks to the Author):

No additional comments

Response: We thank the reviewer for the positive comment.

Reviewer #3 (Remarks to the Author):

The manuscript has been significantly improved and I have no more concerns.

Response: We thank the reviewer for the positive comment.

Reviewer #4 (Remarks to the Author):

Wang et al.

Whole-genome sequencing reveals genomic signatures associated with the inflammatory microenvironments in Chinese NSCLC patients

1. The authors have addressed the comment regarding typos and writing.

Response: We thank the reviewer for the positive comment.

2. The authors have also clarified when WES and WGS data have been used. The manuscript has been modified to address this point.

Response: We thank the reviewer for the positive comment.

3. The review has raised questions regarding the similarity between MS3 and Signature 1B originally reported in Alexandrov et al. (citation #7). The authors have found that even though Signature 1B is similar to MS3 (cosine similarity = 0.92), the original Signature 1B is actually the combination of the other original Signature 1A and Signature 5. According to Alexandrov et al. (2013), "Signatures 1A/B, 2 and 5 were also found in lung adenocarcinoma. Signature 5, but not Signatures 1A/B and 2, also showed a positive correlation between smoking history and mutation contribution ($P = 8.0 \times 10^{-3}$, Supplementary Fig. 96). Thus, in lung cancer, Signature 5, which is characterized predominantly by C>T and T>C mutations, may also be due to tobacco carcinogens." Because MS3 is found in female never-smokers in this study, it may be related to secondary smokers, while MS2 is from primary smokers.

Response: We thank the reviewer for raising this point. Interestingly, the absolute number of mutations characterized by Signature 5 (MS3) increases in smoking lung cancer patients, however, the proportion of this signature is much higher in female never-smokers. The evidence suggests

that these mutations can occur in never-smokers, but accumulate rapidly in smokers. Thus, tobacco carcinogens seem to be an accelerator instead of an inducer. Our results suggested that the inflammation microenvironment can be a candidate inducer because inflammation can happen in every individual when involuntary exposure to carcinogens (secondary smoking as mentioned by the reviewer or other environmental carcinogens) and can be aggravated by direct smoking behavior. Alexandrov et al. also found that “it is also present in nine other cancer types, most of which are not strongly associated with tobacco consumption”, which further supported our conclusions. Thus, our results provide important clues for the underlying mechanisms of never-smoking lung cancer patients.

4. The causal relationship between EGFR mutation and TIL (Reviewer 1’s Comment 3 and Reviewer 3’s Comment 1).

In response to the critiques of Reviewers 1 and 3, the authors demonstrate that the inflammatory microenvironment may occur before occurrence of EGFR mutations (Response Figure 1) using data derived from a recent publication. To more convincingly show that the inflammatory microenvironment preceded EGFR mutation, the authors could also show data demonstrating that the level of infiltrating lymphocytes in AAH are higher in individuals with EGFR mutant ADC as compared to AAH lesions in tumors that do not have EGFR mutation. Inferring temporal relationships and causality from a single point in time remains problematic, as the authors imply.

Response: As suggested by the reviewer, we have compared the level of infiltrating lymphocytes in atypical adenomatous hyperplasia (AAH) lesions from individuals with and without *EGFR* mutation in AD tissues. As expected, infiltrating CD4+ T cells was significantly higher in AAH from individuals with EGFR mutations in tumor tissues (Wilcoxon’s rank sum test $P=0.05$, Response Figure 1). Infiltrating B cells was also higher in AAH from individuals with *EGFR* mutations in tumor tissues (Median: 0.085 vs. 0.083, Response Figure 2), but the difference failed to achieve significance, possibly due to the limited sample size ($n=16$). These results provide additional evidence for the causality between inflammation microenvironments and *EGFR* mutations, but further experimental studies are warranted to directly evaluate the causal relationship.

Response Figure 3. The abundance of B cells and T cells (CD4+) in AAH lesions from individuals with and without *EGFR* mutation in ADC tissues